# A Visual Trajectory-Based Method for Personnel Behavior Recognition in Industrial Scenarios

**DOI:** 10.3390/s25206331

**Published:** 2025-10-14

**Authors:** Houquan Wang, Tao Song, Zhipeng Xu, Songxiao Cao, Bin Zhou, Qing Jiang

**Affiliations:** College of Metrology Measurement and Instrument, China Jiliang University, Hangzhou 310018, China; wanghouquan@cjlu.edu.cn (H.W.); xuzhipeng@cjlu.edu.cn (Z.X.); caosongxiao@cjlu.edu.cn (S.C.); zhoubin@cjlu.edu.cn (B.Z.); 06a0203051@cjlu.edu.cn (Q.J.)

**Keywords:** detection, multiple object tracking, perspective transformation, behavior recognition

## Abstract

Accurate recognition of personnel behavior in industrial environments is essential for asset protection and workplace safety, yet complex environmental conditions pose a significant challenge to its accuracy. This paper presents a novel, lightweight framework to address these issues. We first enhance a YOLOv8n model with Receptive Field Attention Convolution (RFAConv) and Efficient Multi-scale Attention (EMA) mechanisms, achieving a 6.9% increase in AP50 and a 4.2% increase in AP50:95 over the baseline. Continuous motion trajectories are then generated using the BOT-SORT algorithm and geometrically corrected via perspective transformation to produce a high-fidelity bird’s-eye view. Finally, a set of discriminative trajectory features is classified using a Random Forest model, attaining F1-scores exceeding 82% for all behaviors on our proprietary industrial dataset. The proposed framework provides a robust and efficient solution for real-time personnel behavior recognition in challenging industrial settings. Future work will focus on exploring more advanced algorithms and validating the framework’s performance on edge devices.

## 1. Introduction

In the era of Industry, the automated recognition of personnel behavior within environments such as manufacturing plants and logistics centers is critical for ensuring operational safety and enhancing production efficiency [1,2,3]. Traditional monitoring methods that rely on manual inspection are often inadequate for the dynamic and large-scale nature of modern industrial settings due to inherent limitations in response time, coverage, and labor costs. Consequently, vision-based intelligent surveillance systems have emerged as a vital technology [4,5]. By analyzing the motion trajectories of individuals [6], these systems can provide a means of understanding behavior, enabling proactive safety alerts and optimizing management workflows.

However, the direct application of computer vision algorithms to industrial scenarios presents significant and interconnected challenges. These environments are often characterized by complex lighting and substantial variations in target scale. Recent advancements have significantly improved the robustness of object detection. Models from the YOLO (You Only Look Once) series, particularly YOLOv8, have become the benchmark for personnel detection. Many studies [7,8,9,10] focus on adapting such architectures to improve the detection performance of targets in specific contexts. Although these advancements provide a solid foundation for personnel detection, they lack validation in real-world industrial environments, and it remains unclear whether these methods can fulfill practical personnel detection requirements under surveillance conditions. Moreover, a more fundamental issue is the severe perspective distortion introduced by the oblique angles of surveillance cameras [11]. Although perspective transformation is a well-established technique for correcting geometric distortions in other domains [12,13], its necessity as a foundational prerequisite for behavior recognition is often overlooked. This geometric distortion fundamentally corrupts the integrity of motion trajectories in the 2D image plane, invalidating the physical meaning of kinematic features. This methodological gap compromises the validity of analyses that rely on kinematic features derived from uncorrected trajectories.

In the field of behavior recognition, much of the cutting-edge research employs deep learning models [14,15,16], which have achieved superior recognition accuracy on benchmark datasets by modeling spatiotemporal interaction patterns within motion trajectories. However, these models typically rely on vast amounts of training data and are associated with high computational costs, which significantly limit their application in real-world scenarios. In contrast, traditional machine learning algorithms exhibit distinct advantages when processing structured trajectory features. Their requirements for data volume and computational resources are more modest, and both the training and prediction processes are considerably more efficient. The performance bottleneck of such methods, however, lies in the discriminative power of the input features. Therefore, an in-depth exploration of trajectory features is necessary to achieve more reliable behavior recognition.

To bridge this gap, this paper introduces a systematic framework designed to recognize personnel behavior in industrial environments. The contributions are summarized as follows:Development of an integrated processing pipeline tailored for industrial scenarios, which seamlessly combines robust detection, multi-object tracking, perspective correction, and trajectory-feature-based classification into an automated workflow to achieve multi-person behavior recognition.Proposal of an enhanced YOLOv8n personnel detection and localization model:
We integrate RFAConv into the model to more effectively capture information disparities across spatial locations, thus enhancing detection accuracy.The incorporation of the EMA facilitates efficient extraction of small-target features while strengthening the model’s focus on critical attributes.Development of a perspective correction method tailored for industrial settings: The perspective transformation algorithm establishes a mapping relationship from tilted views to top-view space, eliminating interference caused by surveillance camera angles on personnel movement trajectories.Design of discriminative trajectory features: We introduce novel features, including trajectory curvature entropy and primary direction angle, which effectively capture subtle motion patterns.

The remainder of this paper is organized as follows: Section 2 reviews the related work. Section 3 details our proposed methodology. Section 4 describes the experimental setup, including the datasets and evaluation metrics used, and provides a comprehensive analysis of the results. Finally, Section 5 concludes the paper by summarizing its key contributions and discussing potential directions for future research.

## 2. Related Work

### 2.1. Robust Trajectory Acquisition in Industrial Environments

The foundational step for trajectory-based behavior analysis is the acquisition of accurate and continuous motion paths for each individual. This is typically a two-stage process involving object detection followed by multi-object tracking [17]. Industrial environments exacerbate this challenge, presenting a difficult combination of varying lighting conditions, distant personnel appearing as small targets, and real-time, lightweight deployment. Single-stage object detectors have become the standard due to their excellent balance of speed and accuracy, forgoing the computationally expensive region proposal step found in two-stage detectors like the Faster R-CNN series [18]. In particular, models like YOLOv8 provide a mature and efficient framework for such applications [19]. Considering the stringent real-time requirements of industrial settings, we selected YOLOv8n as our base model, distinguished by the fastest inference speed and the smallest parameter count in its series. However, the detection of small objects remains one of the most formidable challenges in computer vision [20], and it is a critical issue in industrial surveillance as personnel are often at a great distance from the cameras [21]. To address this well-documented limitation, a significant body of research has focused on enhancing single-stage detectors specifically for small-object detection. Common strategies involve augmenting multi-scale feature representation and integrating attention mechanisms to guide the model toward salient object features [22,23]. While well-known mechanisms like CBAM and CA enhance feature selectivity, they often apply globally shared attention weights, failing to resolve the misalignment between the convolutional receptive field and the attention’s spatial scope. To address these challenges, our work enhances the YOLOv8n backbone by integrating two complementary attention mechanisms: RFAConv and the EMA module. RFAConv innovatively constrains the attention generation process to the convolutional kernel’s physical receptive field, improving spatial feature capture [24]. The EMA module complements this by employing a parallel cross-spatial learning strategy to efficiently aggregate multi-scale features without channel dimensionality reduction, further enhancing feature discriminability while maintaining a lightweight profile [25]. This deliberate enhancement of a lightweight baseline provides a tailored solution for the specific detection challenges in industrial scenes.

Once personnel are detected, a MOT algorithm is required to associate these detections across frames and maintain consistent identities. Early methods like SORT, which rely solely on motion prediction via a Kalman filter and IoU-based matching, are fast but prone to frequent identity switches during occlusions [26]. More recent trackers like ByteTrack use low-confidence detections to maintain tracks through occlusions, significantly improving trajectory continuity [27]. Building on these advancements, we selected BOT-SORT as our core tracking algorithm. While computationally more intensive than alternatives like ByteTrack, BOT-SORT excels in identity preservation (measured by the IDF1 metric) [28]. These features are critical for maintaining the long, unbroken trajectories and reliable downstream behavior analysis, making it the optimal choice for our application’s end goal.

### 2.2. Geometric Distortion Correction via Perspective Transformation

In most real-world scenarios, surveillance cameras are installed at oblique angles due to physical constraints, rather than in an ideal top-down or orthogonal position [29]. This non-orthogonal viewpoint introduces significant perspective distortion, a geometric effect where objects farther from the camera appear smaller and move more slowly on the image plane, even if their real-world speed is constant. Consequently, the raw 2D trajectories extracted from the image plane are not a faithful representation of the actual motion occurring on the ground. This distortion severely compromises the integrity of spatiotemporal features; To bridge this gap, our work incorporates an explicit perspective correction module. The core of this module is perspective transformation, which remaps the image from its original viewpoint to a synthetic view [30,31]. This transformation establishes a geometric projection relationship between the image plane and the ground plane, effectively removing perspective distortion. Although the efficacy of this transformation is well-established in related domains [32,33,34], its application to the specific problem of personnel trajectory analysis remains notably underexplored. Therefore, the primary contribution of this work is not the invention of the perspective transformation technique, but rather its novel and systematic integration into a pipeline for industrial personnel behavior recognition, which enhances the overall system’s reliability and practical utility.

### 2.3. Personnel Behavior Recognition Model Establishment

The field of Human Activity Recognition offers two primary paradigms: deep learning models and traditional machine learning approaches that rely on engineered features [35,36]. While powerful deep learning approaches like 3D-CNN and LSTM have shown state-of-the-art performance in various domains, their high computational complexity and need for massive labeled datasets make them less suitable for lightweight industrial deployment [16,37]. Consequently, trajectory-based behavior recognition presents a more computationally efficient and practical alternative for industrial settings [14,38]. However, early trajectory-based methods often relied on basic metrics, which are insufficient for distinguishing subtle behavioral differences and are sensitive to noise from tracking inaccuracies [39]. This limitation has driven research towards more discriminative feature engineering, where the goal is to design robust, high-level features that can capture the nuanced signatures of complex motion patterns. For the final classification task, while various models like Support Vector Machines (SVMs) are effective, the Random Forest classifier is particularly well-suited for this application. As an ensemble method, Random Forest demonstrates strong robustness to noisy data, handles high-dimensional feature vectors efficiently, making it an ideal choice for achieving a pragmatic balance between performance and efficiency in real-world scenarios [40,41,42].

## 3. Methodologies

We propose a systematic framework to address the challenges of personnel behavior recognition in industrial environments. Our multi-stage pipeline begins by acquiring robust motion trajectories using an enhanced detection model and a tracking algorithm. A critical perspective transformation module then corrects for geometric distortions. Finally, a classifier utilizes our designed set of high-discriminative trajectory features to perform the final behavior identification. The following subsections will detail each of these core components.

### 3.1. Overall Framework

Figure 1 illustrates the overall framework of the personnel behavior recognition system. The framework integrates four key components: personnel detection and localization via enhanced YOLOv8, personnel tracking using the BOT-SORT algorithm, surveillance footage correction through perspective transformation, and personnel behavior recognition based on trajectory feature engineering and the Random Forest model. The workflow is as follows:Personnel Detection: Video frames are fed into the enhanced YOLOv8 model to localize personnel in an industrial scene.Personnel Tracking: The BOT-SORT tracker associates personnel IDs and generates trajectory position data.Perspective Correction: Surveillance footage undergoes perspective transformation to map raw trajectory coordinates to a standardized bird’s-eye view space.Feature Extraction: Transformed trajectories undergo feature engineering to construct a trajectory feature vector dataset.Behavior Recognition: The Random Forest classifier identifies personnel behaviors based on extracted features.

### 3.2. Behavior Analysis

As illustrated in Figure 2, four behavioral categories are defined based on trajectory features: Normal Walking, Prolonged Stillness, Abnormal Acceleration, and Area Loitering. The trajectory characteristics corresponding to each behavior are defined as follows:Normal Walking: Near-linear trajectory with consistent velocity.Prolonged Stillness: Minimal directional changes and increasing trajectory length over time within specific regions.Abnormal Acceleration: Sudden velocity increase within localized zones.Area Loitering: Extended presence within a confined area with near-zero displacement magnitude.

### 3.3. Improved YOLOv8n

In complex industrial scenarios, the baseline YOLOv8n model encounters two main challenges for personnel detection. The first involves missed detections of distant targets due to inadequate feature representation. Second, localization drift is caused by environmental lighting noise. This limitation stems from the spatial invariance inherent in standard convolution, which inhibits location-sensitive feature extraction, compounded by the lack of cross-channel interaction mechanisms in the Neck network, impairing feature fusion under occlusion. An improved YOLOv8n model is developed for industrial personnel detection. The RFAConv module (denoted by dark blue blocks in Figure 3) is integrated into the Backbone network. By constraining attention weight generation to convolutional kernels’ physical receptive fields, it enables dynamic adaptive enhancement of local features. This design significantly enhances structural representation capability, boosting feature response intensity for distant personnel. Additionally, the EMA module (marked by orange blocks in Figure 3) is embedded in the Neck network. Its parallel cross-channel interaction mechanism compensates for information loss from occlusion and strengthens the semantic fusion of multi-scale features. The overall architecture of the improved YOLOv8 is illustrated in Figure 3.

#### 3.3.1. Incorporation of the RFAConv

To enhance the feature extraction capabilities of our network, we replace a standard convolutional block with the RFAConv. The primary limitation of standard convolution is its static parameter-sharing mechanism, which applies the same filter across all spatial locations. RFAConv addresses this by dynamically generating unique attention weights for each receptive field, enabling the network to focus on the most information-rich regions and improve detection accuracy for objects of varying sizes and shapes.

As illustrated in Figure 4, the RFAConv implements this adaptive mechanism through a dual-branch architecture that processes the input feature map in parallel:This branch generates attention maps by first using average pooling to aggregate global information, followed by 1 × 1 convolutions for efficient channel interaction. A Softmax function then produces the final weights, which highlight the most critical feature locations within the receptive field.The Spatial Feature Branch: Operating concurrently, this branch uses grouped convolutions to extract a rich set of spatial feature maps from the same input features. Each group of convolutions generates a distinct feature map corresponding to a specific part of the receptive field.

The outputs of these two branches are then fused via element-wise multiplication. This operation re-weights the spatial features using the learned attention maps, effectively amplifying salient information and suppressing irrelevant details. The resulting feature map is then processed by a final standard convolution. This entire process allows the feature extraction to be adaptively tailored for each specific region of the input data, significantly boosting the model’s overall detection performance.(1)F=Softmax(gi×i(AvgPool(X)))×ReLU(Norm(gk×k(X)))=Arf×Frf

$g^{i\times i}$ represents a group convolution of size i×i, k represents the convolution kernel size, Norm represents normalization, X represents input feature maps, and F is obtained by multiplying the attention map $A_{rf}$ with transformed receptive field spatial features $F_{rf}$. To enhance the model’s feature extraction capabilities while managing inference time, we replace the standard convolutions in the YOLOv8 backbone with RFAConv, retaining only the original first layer to ensure computational efficiency.

#### 3.3.2. Incorporation of the EMA

The EMA integrates multi-scale feature extraction with cross-space learning to precisely capture target features across scales. Its core design employs parallel processing of multi-scale information and dynamic weight allocation to reinforce critical features, enhancing model accuracy while maintaining low computational overhead. The structure is shown in Figure 5.

EMA adopts cross-space information aggregation along different spatial dimensions to achieve richer feature integration. Specifically, two tensors from the 1×1 and 3×3 branch outputs are introduced. A 2D global average pooling operation then encodes global spatial information from the 1 × 1 branch output, while the minimal branch output is reshaped to corresponding dimensions prior to channel feature fusion and activation. Let $x_{c}$ denote the input features at the c-th channel. The 2D global pooling operation is formulated as follows:(2)$$z_{c}(W)=\frac{1}{H\times W}\sum_{j}^{H}\sum_{i}^{W}x_{c}(i,j)$$

It encodes global information and models long-range dependencies. Softmax is applied to 2D global average pooling outputs for linear adaptation. Matrix dot products of parallel-processed outputs generate the first spatial attention map, aggregating multi-scale spatial information. Similarly, 3 × 3 and 1 × 1 branches undergo global pooling, with 1 × 1 outputs reshaped for joint channel activation to produce the second precise spatial attention map. Fused attention weights processed through Sigmoid yield grouped feature maps, capturing pixel-wise relationships and global context.

The final output dimension of the model is consistent with X, ensuring computational efficiency and compatibility with modern neural network architectures. Thus, EMA is integrated into key feature fusion nodes of the Neck network: inserted after shallow feature and deep semantic fusion to enhance the detail characteristics of small targets using its multi-scale perception capability; deployed at intermediate feature fusion nodes to balance global semantics and local details via cross-space interaction; and incorporated after deep feature fusion to reduce redundancy in high-level semantic features through its channel grouping strategy. This refinement significantly enhances personnel detection accuracy in various scenarios without substantially increasing computational overhead.

### 3.4. Perspective Transformation

Due to the perspective effect of surveillance cameras, the rate of trajectory length increase decreases as personnel move away from the camera and increases as they approach it. This results in distortion of personnel trajectories in the raw imaging plane, thereby interfering with subsequent behavior recognition. Therefore, perspective transformation algorithms are employed to eliminate such interference. Perspective transformation is a technique that projects skewed or distorted images to a canonical viewpoint using geometric transformation, with its core concept centered on establishing spatial mapping relationships between source and target images. This process is achieved through a 3×3 homography matrix H [43], which describes linear transformation relationships of pixel points in homogeneous coordinates. Specifically, given a point (x,y) in the source image and its corresponding point $\left(x^{\prime },y^{\prime }\right)$ in the target image, the transformation is formulated as follows:(3)$$s\left[\begin{array}{c} x^{\prime }\\ y^{\prime }\\ 1 \end{array}\right]=H\left[\begin{array}{c} x\\ y\\ 1 \end{array}\right]=\left[\begin{array}{ccc} h_{11} & h_{12} & h_{13}\\ h_{21} & h_{22} & h_{23}\\ h_{31} & h_{32} & h_{33} \end{array}\right]\left[\begin{array}{c} x\\ y\\ 1 \end{array}\right]$$
where s denotes the scale factor. The matrix H can be uniquely determined using at least four pairs of non-collinear corresponding points. Perspective transformation is applied to surveillance footage as depicted in Figure 6. This process maps selected regions onto a bird’s-eye view projection, thereby reconstructing true spatial trajectories of personnel movement in physical scenes.

Performing perspective transformation requires five sequential steps:

Feature point calibration: The four vertices of the rectification area in the surveillance image are located through annotation.The target image dimensions are defined based on the scale ratio of the region of interest.Execute matrix computation and transformation: Compute the matrix H using the DLT [44] with four pairs of source-target point correspondences.For each pixel $\left(x^{\prime },y^{\prime }\right)$ in the target image, inversely map to source coordinates (x,y) via $H^{-1}$.The pixel value at (x,y) is computed via interpolation algorithms and populated into the target image.

Subsequent experiments demonstrate that perspective transformation effectively mitigates interference in personnel behavior recognition caused by camera perspective effects.

### 3.5. Trajectory Feature Engineering

This study designs discriminative trajectory features to represent motion patterns, providing reliable training data for subsequent behavior recognition models. Specifically, four core trajectory features are constructed:

Trajectory Curvature Entropy

Curvature quantifies the bending degree of a curve. For each discrete trajectory point $p_{i}=\left(x_{i},y_{i}\right)$, let $k_{i}$ denote its curvature, computed through derivatives of parametric equations from three adjacent points:(4)$$k_{i}=\frac{{\dot{x}}_{i}{\ddot{y}}_{i}-{\dot{y}}_{i}{\ddot{x}}_{i}}{{\left({\dot{x}}_{i}^{2}+{\dot{y}}_{i}^{2}\right)}^{3/2}}$$

Entropy quantifies the disorder degree of data or signals [45]. For discrete points along a trajectory curve, directional variation magnitude can be characterized using entropy. This work employs trajectory curvature entropy computation to characterize the disorder level of personnel trajectories. The trajectory curvature entropy is computed as follows:(5)$$E_{dir}=-\sum_{i=1}^{n}H_{i}(dir)log_{2}H_{i}\left(dir\right)$$
where $E_{dir}$ denotes the trajectory curvature entropy, and $H_{i}(dir)$ represents the probability distribution of directional variation magnitudes. Analysis of this entropy formulation reveals that $E_{dir}$ exhibits a strong correlation with personnel movement directionality: More concentrated directional changes yield lower entropy values, while more dispersed directional variations result in higher entropy.

Primary Direction Angle

The planar angular range [0°, 360°) is uniformly partitioned into B equal-width bins. Let $N_{\theta }$ denote the frequency distribution of trajectory points’ direction angles across bins, where $N_{A}$ is the total frequency count, and $N_{max}$ is the maximum frequency within any angular bin. If there exists a bin satisfying $p=\frac{N_{max}}{N_{A}}>\sigma$ (σ is a preset threshold), then the central angle of that bin is assigned as the primary direction angle θ. Otherwise, select the minimal number c of $N_{\theta_{i}}$ satisfying $\sum_{i=1}^{c}\frac{N_{\theta_{i}}}{N_{A}}>\sigma$. The average direction angle $\theta_{avg}$ is then designated as the primary direction angle θ, computed as follows:(6)$$\theta_{avg}=\frac{\sum_{i=1}^{k}\theta_{i}}{c}$$
where $\theta_{i}$ denotes the angular value corresponding to the i-th bin. Direction angle distributions of Normal Walking trajectories exhibit high concentration, whereas Area Loitering induces dispersed direction angle distributions. The primary direction angle θ characterizes the dominant trend of personnel motion direction.

Motion Velocity and Directional Anomaly

From trajectory points (x,y), first-order differences (dx,dy) along the x-axis and y-axis are computed, implicitly representing velocity information components. Furthermore, second-order differences $\left(\left|d^{2}x\right|,\left|d^{2}y\right|\right)$ are introduced to characterize velocity variations, thereby refining the characterization of velocity dynamics. For the x-axis direction, the computational formulas are as follows:(7)$$\left\{\begin{array}{c} dx=x_{i}-x_{i-1}\\ d^{2}x=x_{i}-2x_{i-1}+x_{i-2} \end{array}\right.$$

Normal Walking trajectories manifest as near-linear paths without frequent directional changes, whereas alternating increases and decreases in consecutive x/y coordinates suggest directional instability. For the x-axis, if positions exhibiting coordinate increases exceed half of the total points, then points with coordinate decreases are defined as anomalous. The sum of the differences of the position points at the reduced positions is taken as the abnormal values in the x-direction. Analogous computation applies to the y-axis. The pair $\left(sumdx,sumdy\right)$ quantifies directional anomalies, with sumdx formulated as follows:(8)$$sumdx=\left\{\begin{array}{c} \sum_{dx\geq 0}dx,{\;N}_{dx}^{-}>\frac{n}{2}\\ \sum_{dx<0}dx,\;{\;N}_{dx}^{+}\geq \frac{n}{2} \end{array}\right.$$
where n denotes the total number of position points, and $N_{dx}^{-}$ and $N_{dx}^{+}$ represent the sums of position points exhibiting coordinate decreases and increases, respectively.

Path Length and Displacement

In consecutive video frames, the initial frame where a person is first detected serves as the starting point. Compute the displacement ΔS and path length S of the person’s movement from the starting frame to the current frame. To further characterize trajectory properties, calculate the ratio r of displacement to path length, defined as r=ΔS/S. As r approaches 1, the person’s trajectory increasingly approximates a straight line.

Based on the four primary features, we select nine parameters: trajectory curvature entropy $E_{dir}$, primary direction angle θ, velocity variation $\left(\left|d^{2}x\right|,\left|d^{2}y\right|\right)$, anomaly in the x/y-direction $\left(sumdx,sumdy\right)$, displacement ΔS, path length S, and ratio r. All trajectory features are integrated into a feature vector as formulated below:(9)$$F=\left[E_{dir},\;\theta ,\left|d^{2}x\right|,\left|d^{2}y\right|,sumdx,sumdy,\Delta S,S,r\right]$$

## 4. Experiments and Analysis

### 4.1. Experimental Settings

#### 4.1.1. Dataset

For personnel detection tasks, we constructed a specialized industrial personnel dataset. The dataset was curated from surveillance footage across multiple real-world industrial scenarios, systematically sampled by extracting one frame per minute, followed by a rigorous manual screening process.

As illustrated in Figure 7, the dataset is designed to comprehensively encapsulate the complexity and diversity of industrial environments. It features a wide range of challenging conditions, including (a) varied illumination (both full and partial illumination), (b) significant scale variations of personnel (both distant and close-range targets), and (c) diverse working scenarios, thereby ensuring the generalization capability of the trained model and the validity of its evaluation. The dataset was partitioned into a training set (3165 images) and a validation set (1356 images). To facilitate a targeted evaluation of the model’s robustness, three distinct test sets were also established: a low-light conditions set (205 images), a distant-target set (212 images), and a general-purpose set (792 images). Figure 8 presents a statistical analysis of the annotations, where all coordinates are normalized (calculated as the ratio of the object’s dimensions to the image’s dimensions). The bounding box size distribution in Figure 8a confirms the significant multi-scale nature of personnel targets within the dataset. The visualization of bounding box centers in Figure 8b reveals a broad spatial distribution, with concentrations in the lower-middle and peripheral regions; this positional variance provides the rationale for integrating RFAConv to enhance the model’s spatial feature adaptation. Furthermore, the scatter plot of bounding box dimensions in Figure 8c shows a high density of instances in the lower-left quadrant, indicating a significant prevalence of small-scale targets. This characteristic was the key motivation for incorporating the EMA attention mechanism to improve the model’s sensitivity to small object detection.

For evaluating object tracking algorithms, we selected three representative video clips (totaling 1251 frames) from industrial sites, containing scenarios with object occlusion, varying distances, and complex conditions.

For evaluating personnel behavior recognition algorithms, videos with rich behavioral patterns were extracted from Figure 8. Using a sliding window approach, we extracted trajectory samples for all personnel in the videos. The trajectory sampling window spanned 250 frames with a step size of 75 frames, yielding 1522 trajectory samples. Each trajectory was annotated with behavior labels: Normal Walking 0, Area Loitering 1, Abnormal Acceleration 2, and Prolonged Stillness 3. The dataset was split into training and test sets at an 8:2 ratio, allocating 1217 trajectories for Random Forest training and 305 for testing.

#### 4.1.2. Implementation

All experiments were conducted on a workstation running a Windows 10 operating system. The hardware platform consisted of an Intel Core i7-11700F processor, 16 GB of system RAM, and an NVIDIA GeForce RTX 3060 GPU with 12 GB of dedicated video memory. The software environment was built on Python 3.12.3, utilizing the PyTorch 2.5.1 deep learning framework with CUDA 12.1 for GPU acceleration. In alignment with our research goals emphasizing real-time operation and a lightweight model architecture, YOLOv8n was selected as the baseline for all experiments. To enhance the robustness of our model and mitigate the risk of overfitting on the custom industrial dataset, a variety of data augmentation techniques were applied during the training phase. The primary hyperparameters used for training are detailed below:
Input image size: 640 × 640;Epoch: 150;Initial learning rate: 0.01;Weight decay: 0.05;Batch size: 8;Optimizer: SGD;Patience: 20;Data enhancement strategy: mosaic, flip, scale, translate, hsv.


### 4.2. Evaluation Metrics

#### 4.2.1. Evaluation Metrics for Detection

Given the exclusive focus on personnel detection, this study employs average precision (AP) as the primary evaluation metric for the enhanced YOLOv8n model. AP quantifies detection accuracy as the area under the precision–recall (PR) curve, formally expressed as follows:(10)$$AP=\int_{0}^{1}p\left(r\right)dr$$
where $p\left(r\right)$ represents the P-R curve. Higher AP values indicate superior capability in identifying targets. Building upon this foundation, AP50 evaluates detection reliability at a 0.5 Intersection-over-Union (IoU) threshold, AP50:95 extends this evaluation by averaging AP across IoU thresholds from 0.5 to 0.95. These metrics comprehensively demonstrate the model’s proficiency in reliably detecting and precisely locating personnel. Meanwhile, Model Size, GFLOPs, and FPS are utilized to assess computational efficiency and practical deployment potential.

#### 4.2.2. Evaluation Metrics for Tracking

For object tracking, we employ the current mainstream evaluation metrics: multi-object tracking precision (MOTP) and multi-object tracking accuracy (MOTA). MOTP quantifies the localization accuracy of tracked targets across frames, with lower values indicating superior performance, computed as follows:(11)$$MOTP=\frac{\sum_{t=1}^{T}\sum_{i=1}^{c\left(t\right)}d\left(t,i\right)}{\sum_{t=1}^{T}c\left(t\right)}$$
where $d\left(t,i\right)$ denotes the positional error of the i-th matched target in frame t, $c\left(t\right)$ is the count of successfully matched targets in frame t, and T is the total number of frames.

MOTA measures overall tracker performance, where higher values denote better system effectiveness. This metric encompasses detection accuracy, tracking precision, and identity consistency through the formulation:(12)$$MOTA=1-\frac{\sum_{t}\left(FN_{t}+FP_{t}+IDSW_{t}\right)}{\sum_{t}g_{t}}$$
where $FN_{t}$ is the count of missed ground-truth targets in frame t, $FP_{t}$ represents false positive detections in frame t, $IDSW_{t}$ indicates identity switches occurring in frame t, and $g_{t}$ is the total ground-truth targets in frame t.

#### 4.2.3. Evaluation Metrics for Personnel Behavior Recognition

For personnel behavior recognition, the primary performance metrics comprise ${Precision}_{i}$, ${Recall}_{i}$, and ${F1-Score}_{i}$ corresponding to each behavior category. The computational formulas are defined as follows:(13)$${Precision}_{i}=\frac{TP_{i}}{TP_{i}+FP_{i}}$$(14)$${Recall}_{i}=\frac{TP_{i}}{TP_{i}+FN_{i}}$$(15)$${F1-Score}_{i}=\frac{2\cdot TP_{i}}{2\cdot TP_{i}+FP_{i}+FN_{i}}$$
where subscript i denotes distinct behavior categories (i=0,1,2,3)

### 4.3. Object Detection Experiments

#### 4.3.1. Comparative Experiments of Different Models

To contextualize our model’s performance, we conducted a comparative analysis against several mainstream detectors, benchmarking them on accuracy, complexity, and speed (Table 1). The results show that architectures like Faster-RCNN are unsuitable for this real-time application due to their large size and slow inference. Within the modern YOLO family, a clear trade-off exists: lightweight models (e.g., YOLOv5-n, YOLOv8-n) offer high FPS but lower accuracy, while larger variants (e.g., YOLOv8-s) improve accuracy at the cost of significantly increased complexity and reduced speed.

In the results of our comparative experiments, our proposed model breaks this trade-off, establishing a superior balance. It achieves the highest accuracy of all tested models, reaching 78.2% in AP50 and 44.8% in AP50:95, while maintaining a highly compact architecture (6.0 MB size, 8.4 GFLOPs) comparable to the baseline YOLOv8-n. Although not the fastest, its 201 FPS is more than sufficient for real-time monitoring. These results validate that our enhancements deliver an accuracy advantage without sacrificing the lightweight efficiency essential for practical edge deployment.

#### 4.3.2. Ablation Study

To systematically validate the contribution of each proposed architectural improvement, we conducted a series of ablation experiments on our custom personnel dataset. As detailed in Table 2, the study was designed to isolate the impact of each module on the baseline model’s performance. The integration of the RFAConv module, which enhances the capture of subtle spatial information via adaptive receptive field features, yielded a significant performance increase, boosting the AP50 by 4.5% and the AP50:95 by 3.6% over the baseline. Similarly, the EMA module, designed to precisely capture multi-scale features with minimal computational overhead, independently improved AP50 by 2.8% and AP50:95 by 2.9%.

When both modules were integrated, their complementary functions produced a synergistic effect, with the completely improved model achieving final increases of 6.9% in AP50 and 4.2% in AP50:95. Crucially, these accuracy gains were realized while maintaining practical efficiency; the model’s size, FPS, and GFLOPs remain well within the constraints required for lightweight deployment. These results empirically confirm that our enhancements work in concert to significantly improve object detection accuracy without compromising performance.

#### 4.3.3. Visualization Analysis

To address the inherent lack of interpretability in deep learning models and to visually validate the efficacy of our architectural enhancements, we employed Grad-CAM [46] for visual analysis. This technique highlights salient regions critical for model predictions, generating visual explanations for deep neural network decisions. Results are presented in Figure 9. The analysis reveals a critical limitation in the baseline model. As depicted in Figure 9b, its heatmaps exhibit diffuse and weak activation on distant, small-scale targets, in some cases failing to focus on them entirely. This lack of attention correlates directly with its lower detection accuracy for such objects. In stark contrast, our optimized YOLOv8n model demonstrates a markedly improved ability to localize these challenging targets. The corresponding heatmaps in Figure 9c show highly concentrated and intense activation regions. This visual evidence provides strong qualitative corroboration for our quantitative results, illustrating that our enhanced model effectively focuses its attention on critical objects, which directly translates to superior detection performance and improved generalization.

#### 4.3.4. Performance on Environment-Specific Test

To verify the robustness of our proposed model, we conducted a targeted evaluation on three distinct test sets: (a) a low-light condition set, (b) a distant-target set, and (c) a general-purpose set. This analysis was designed to empirically measure the model’s performance under the specific, challenging conditions it was designed to address. The results, summarized in Table 3, confirm the targeted benefits of our architectural modifications.

In the low-light condition (a), our model achieved a marked improvement, outperforming the baseline by 8.0% in AP50 and 4.8% in AP50:95, demonstrating the model’s ability to discern salient features in low-visibility backgrounds. The most substantial gains were observed on the distant-target set (b); our model increased the AP50 by 10.9% and the more stringent AP50:95 metric by 8.8%. This directly validates the effectiveness of our enhancements for detecting and precisely localizing small-scale objects. The performance on the general-purpose set (c) reaffirmed the robust, all-around benefit of our enhancements. These findings confirm that our model is not only more accurate overall but also significantly more reliable in the diverse and challenging environments characteristic of industrial settings.

### 4.4. Experiments on Tracking

For the object tracking phase, we utilized a ReID model pre-trained on the extensive Market-1501 dataset [47]. Given that this dataset contains a vast array of pedestrian images captured under diverse camera perspectives and lighting conditions, providing a rich feature set for distinguishing individuals, retraining the ReID model was deemed unnecessary for this study. To construct a dedicated tracking test set, three challenging video sequences, totaling 1251 frames and featuring scenarios with target occlusion and significant distance variations, were selected and meticulously annotated using X-AnyLabeling [48]. Detailed experimental scenarios and tracking performance metrics are presented in Table 4.

The experimental results, detailed in Table 4, demonstrate the tracker’s robust performance under typical industrial conditions. In Scenes 1 and 2, BOT-SORT maintained superior tracking efficacy, even when targets were distant or experienced partial occlusion and overlap. In contrast, a moderate degradation in tracking efficacy was observed in Scene 3, primarily due to instances of prolonged target concealment by large, static obstructions, which led to missed detections by the upstream model (as illustrated in Figure 10c).

In summary, the evaluation confirms the algorithm’s fundamental capability for personnel tracking in industrial environments. Across the different scenarios, the tracker achieved minimum MOTA and MOTP values of 71.92% and 13.90%, respectively. Furthermore, the integrated detection and tracking pipeline consistently operates at a speed exceeding 56 FPS, satisfying the real-time requirements for practical monitoring applications.

### 4.5. Perspective Transformation Experiment

To validate the critical role of perspective transformation in personnel behavior recognition, comparative experiments were conducted. We conducted tests on a representative video segment from the same scene under stable detection frame rates. Corresponding trajectory plots were placed on the left side of the interactive interface for intuitive visualization of motion patterns. As shown in Figure 11, without perspective transformation (Figure 11a), the classifier mistakenly identified Normal Walking as an Abnormal Acceleration state. This misjudgment occurs due to camera perspective interference: as personnel approach the camera, the trajectory exhibits a progressive change in length, resulting in densely clustered distant points and sparse proximal points, which distorts the overall geometry of the trajectory and leads to incorrect behavior classification. In contrast, when perspective transformation was applied (Figure 11b), the trajectory demonstrated a uniform increase in length over time with evenly distributed points, correctly identifying Normal Walking behavior. These comparative results fully demonstrate the essential role of perspective transformation in enhancing the accuracy of behavior recognition.

### 4.6. Personnel Behavior Recognition Experiment

#### 4.6.1. Classifier Optimization and Feature Validation

The trajectory database contains 1522 samples across four behaviors: Normal Walking (474), Area Loitering (485), Abnormal Acceleration (145), and Prolonged Stillness (418). The dataset was first partitioned into a training set (1217 samples) and a test set (305 samples) at an 8:2 ratio. An initial analysis revealed a significant class imbalance, with the Abnormal Acceleration class being underrepresented (Figure 12, left).

To mitigate the risk of classifier bias, we first applied the Synthetic Minority Oversampling Technique (SMOTE) to the training data. This process generated synthetic samples for the minority class, creating a more balanced class distribution for model training (Figure 12, right). With the augmented training data, we then optimized the Random Forest classifier. The model’s hyperparameters were determined through a systematic Grid Search with cross-validation performed on the trajectory data. This process identified the optimal parameter combination for our specific dataset, with the best-performing parameters detailed below:

class_weight = “balanced”;max_depth = 15;min_samples_leaf = 1;min_samples_split = 2;n_estimators = 150.

To validate the low inter-feature correlation and non-redundancy of the designed trajectory features, we generated a feature importance histogram (Figure 13) derived from the Random Forest model alongside a Pearson correlation coefficient heatmap (Figure 14).

Analysis of Figure 13 reveals that among the engineered trajectory features, trajectory curvature entropy exhibits the predominant contribution, while the remaining features demonstrate comparable importance levels. This finding substantiates the discriminative capability of our trajectory feature set for distinguishing personnel behavioral patterns.

As evidenced in Figure 14, the majority of feature pairs maintain correlation coefficients below 0.8, with the exception of path length versus displacement. This statistical evidence confirms the absence of significant inter-feature correlations, thereby establishing the designed trajectory features as non-redundant.

#### 4.6.2. Experimental Results and Analysis

The optimized Random Forest classifier was trained on the augmented training data and evaluated on the original, unseen test set. The classification performance is detailed in the confusion matrix (Figure 15) and summarized in Table 5. The results indicate that the model achieves high accuracy across all four behaviors, with precision, recall, and F1-score values consistently exceeding 82%. The successful recognition of the ‘Abnormal Acceleration’ class, in particular, validates the effectiveness of our combined data augmentation and model optimization strategies.

As shown in Figure 16, the recognition process for four behaviors is demonstrated. The overall performance demonstrates that our trajectory-based behavior recognition model is robust and reliable for deployment in complex industrial environments.

These results highlight the solid performance of the proposed trajectory-feature-based method for personnel behavior recognition. While deep learning architectures offer the benefit of automatic feature learning [16,37], they are often computationally intensive, which limits their potential for deployment on resource-constrained edge devices. The proposed lightweight framework, in contrast, provides superiority in computational efficiency and model simplicity. However, extending the system to recognize entirely new, unforeseen behaviors may require additional feature engineering, sacrificing the flexibility of end-to-end deep learning models. Despite this limitation, the high effectiveness demonstrated in the experiments confirms that the proposed method is a robust and reliable solution that meets the demands of industrial scenarios.

## 5. Conclusions

This paper proposes a novel method for personnel behavior recognition, which comprises five structured stages. First, RFAConv and an EMA are integrated into the YOLOv8 model, enhancing detection accuracy with a 6.9% increase in AP50 and a 4.2% increase in AP50:95 over the baseline. Second, the BOT-SORT algorithm is utilized for personnel tracking. Then, a perspective transformation algorithm is applied to correct motion trajectories distorted by surveillance camera views. Furthermore, discriminative dynamic trajectory features are innovatively designed to overcome the limitations of conventional trajectory representations. Finally, a Random Forest classifier is adopted to recognize personnel behaviors, achieving F1-scores exceeding 80% across all four behavioral categories. Experimental results indicate that the proposed method demonstrates strong performance in personnel behavior recognition. Despite this strong performance, we acknowledge several limitations. The tracking module’s efficacy degrades under conditions of prolonged, complete occlusion, a persistent challenge in the tracking-by-detection paradigm. Furthermore, extending the system to recognize entirely new, unforeseen behaviors may require additional feature engineering, sacrificing the flexibility of end-to-end deep learning models.

Future work will focus on two key directions. First, we will explore more advanced object tracking algorithms to improve robustness against long-term occlusions. Second, a critical next step involves deploying and validating the entire framework on resource-constrained edge devices to rigorously evaluate its real-world performance and computational efficiency. This approach is expected to provide reliable technical support for industrial safety surveillance systems.

## Figures and Tables

**Figure 1 sensors-25-06331-f001:**
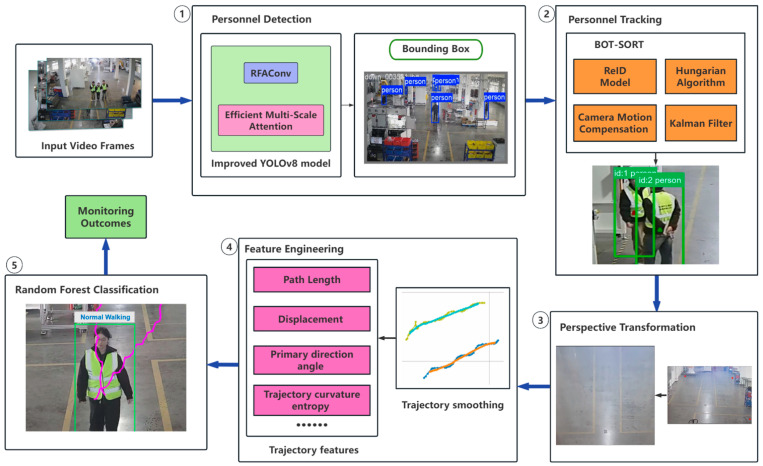
The overall framework of the proposed personnel behavior recognition system, illustrating five core processing steps.

**Figure 2 sensors-25-06331-f002:**
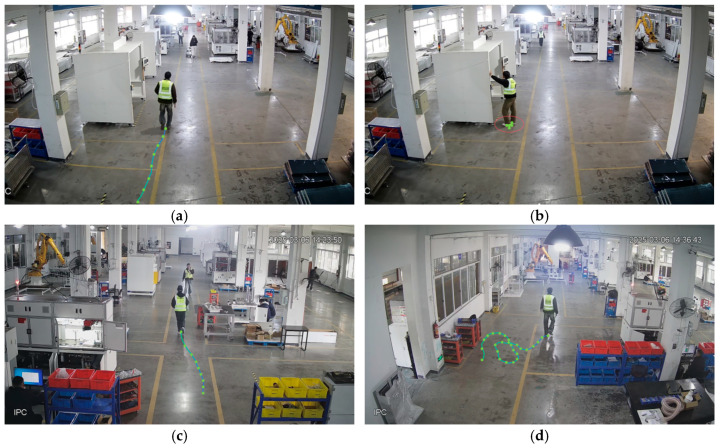
Trajectory diagrams of various personnel behaviors: (**a**) Normal Walking; (**b**) Prolonged Stillness; (**c**) Abnormal Acceleration; (**d**) Area Loitering.

**Figure 3 sensors-25-06331-f003:**
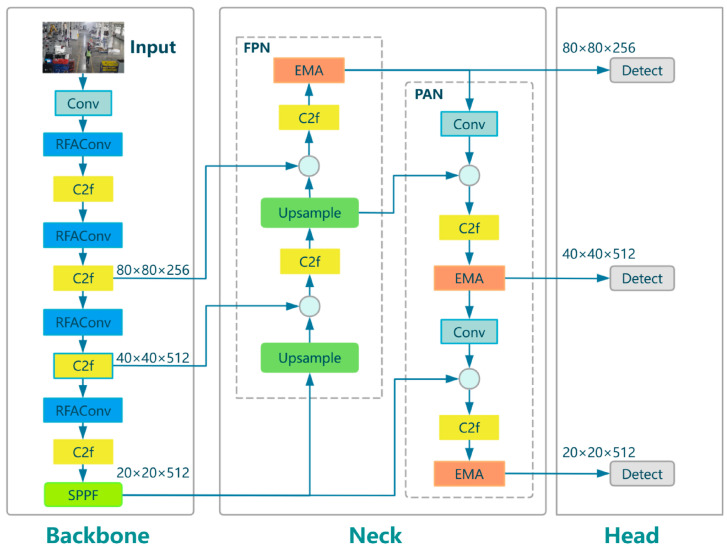
The overall architecture of our improved YOLOv8 network is described in detail.

**Figure 4 sensors-25-06331-f004:**
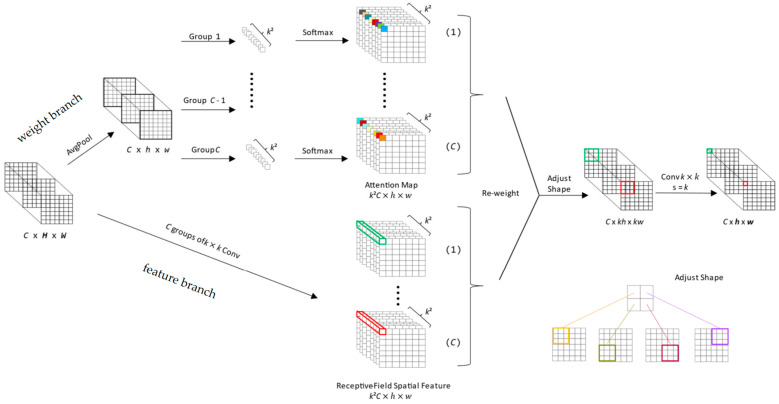
The structure of RFAConv. The input is processed by two branches to generate an attention map and a spatial feature map, which are then fused to achieve feature extraction that emulates non-shared parameters.

**Figure 5 sensors-25-06331-f005:**
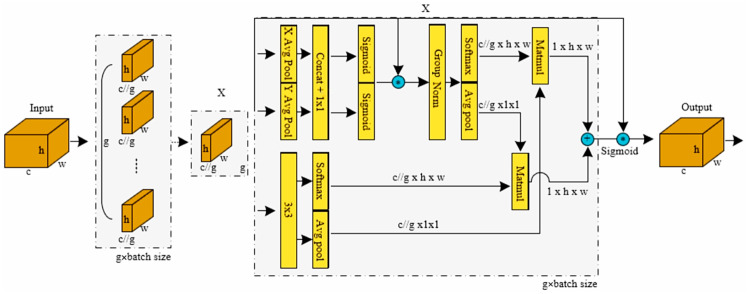
Detailed structure diagram of EMA, where g denotes the number of segmentation groups, and XAvgPool/YAvgPool represent horizontal and vertical global pooling, respectively.

**Figure 6 sensors-25-06331-f006:**
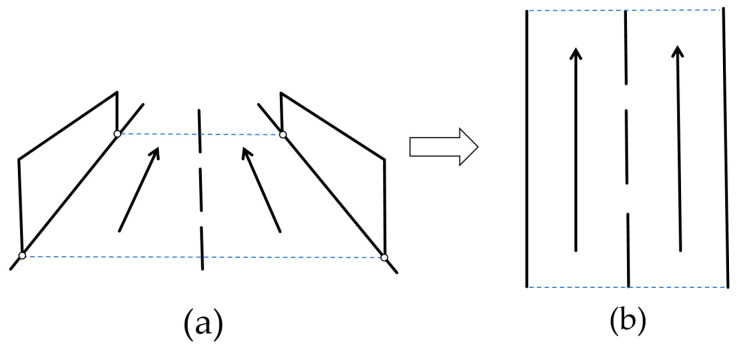
Perspective transformation schematic: (**a**) original camera view; (**b**) top-down bird’s-eye view after transformation.

**Figure 7 sensors-25-06331-f007:**
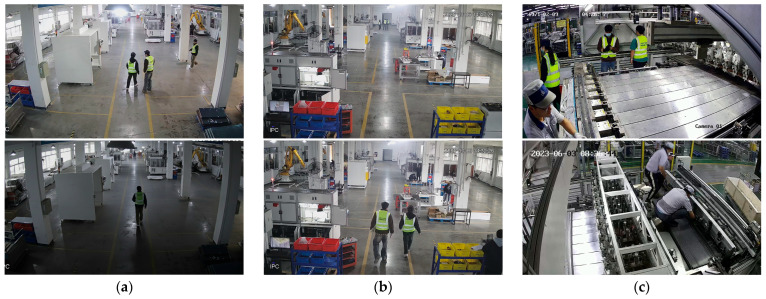
Sample images from the curated industrial dataset: (**a**) varied illumination conditions; (**b**) significant scale variation in personnel; (**c**) diverse industrial scenes.

**Figure 8 sensors-25-06331-f008:**
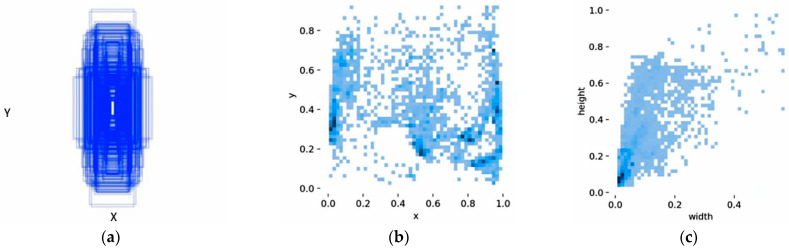
Statistical information of the object detection dataset: (**a**) size distribution of object bounding boxes; (**b**) scatter plot of bounding box width versus height; (**c**) spatial distribution of object locations.

**Figure 9 sensors-25-06331-f009:**
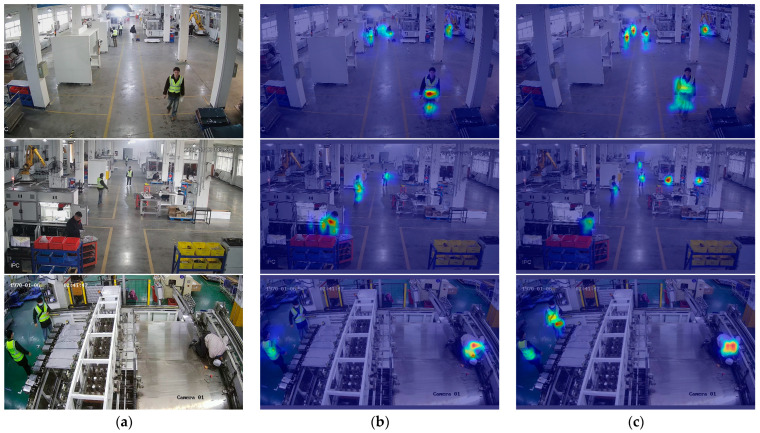
Comparison of heatmaps between the baseline and the model: (**a**) original image; (**b**) Baseline image; (**c**) our model image.

**Figure 10 sensors-25-06331-f010:**
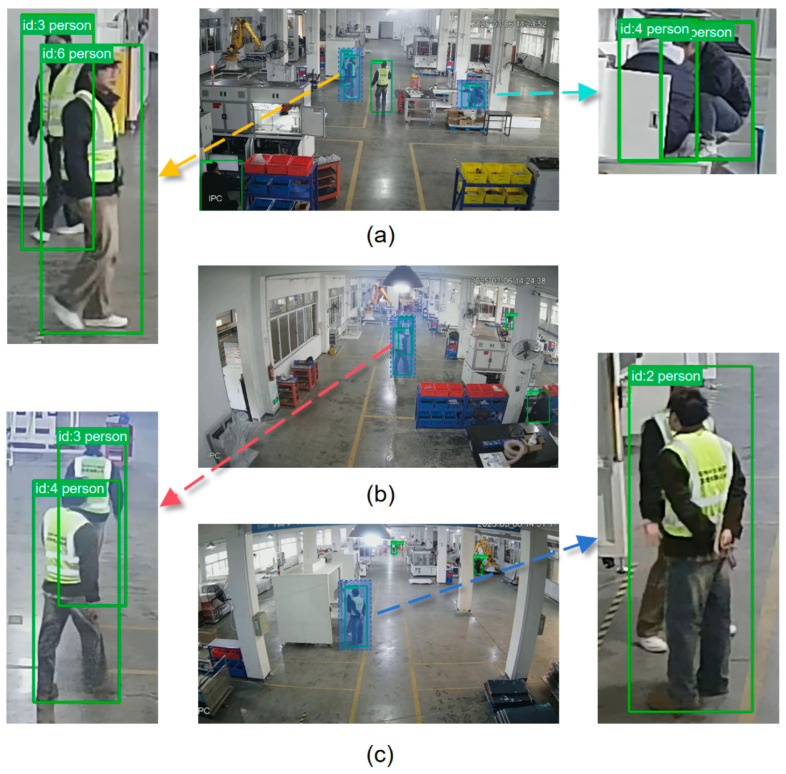
Visualization of three tracking scenarios: (**a**–**c**) Scenes 1–3; (**a**,**b**) stable tracking instances with partial target overlap; (**c**) tracking failure instance caused by extensive occlusion resulting in missed detection.

**Figure 11 sensors-25-06331-f011:**
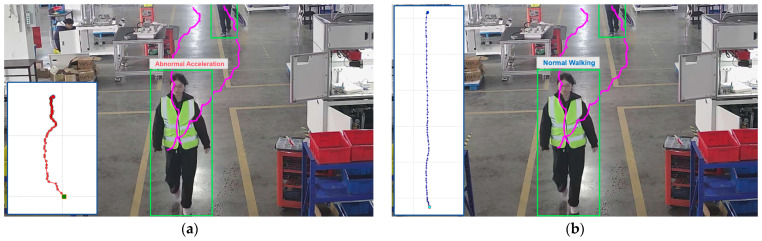
Perspective transformation efficacy comparison: (**a**) erroneous behavior identification without transformation; (**b**) correct behavior identification with transformation.

**Figure 12 sensors-25-06331-f012:**
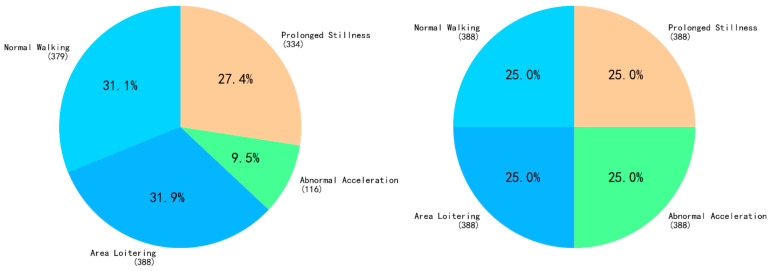
Pie charts showing class distribution before and after SMOTE.

**Figure 13 sensors-25-06331-f013:**
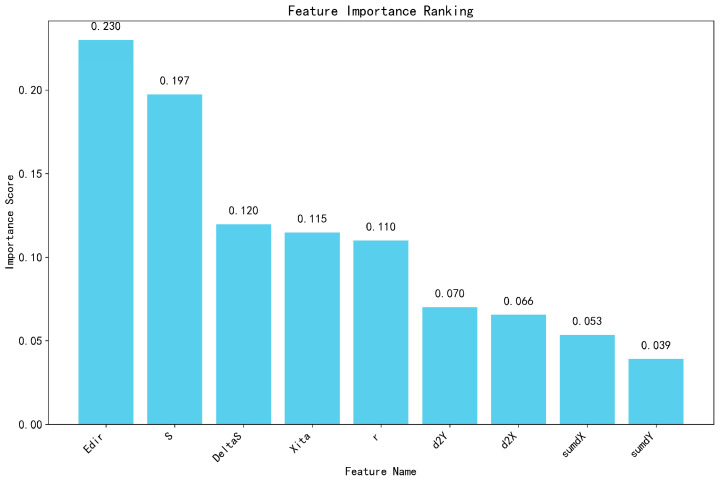
Importance ranking of trajectory features.

**Figure 14 sensors-25-06331-f014:**
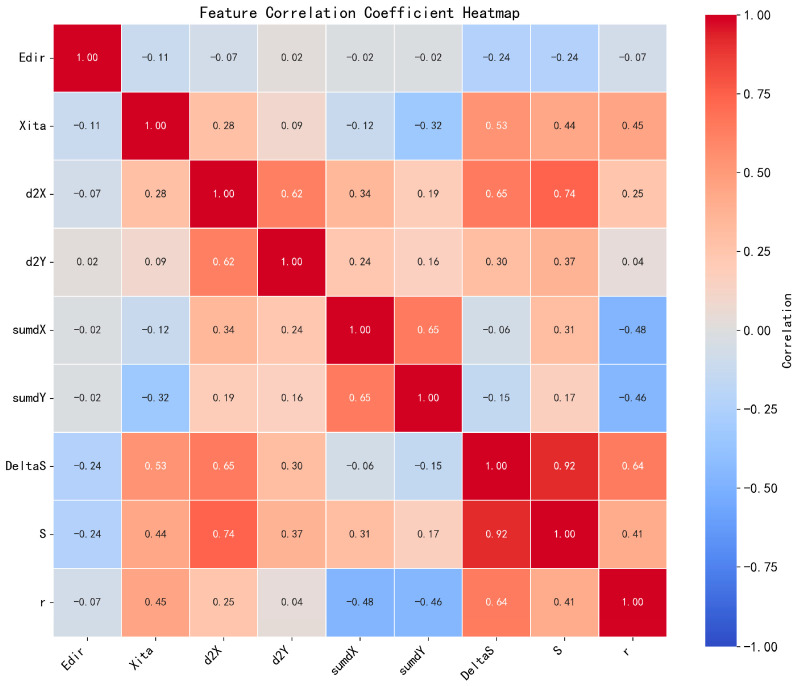
Pearson correlation heatmap of trajectory features.

**Figure 15 sensors-25-06331-f015:**
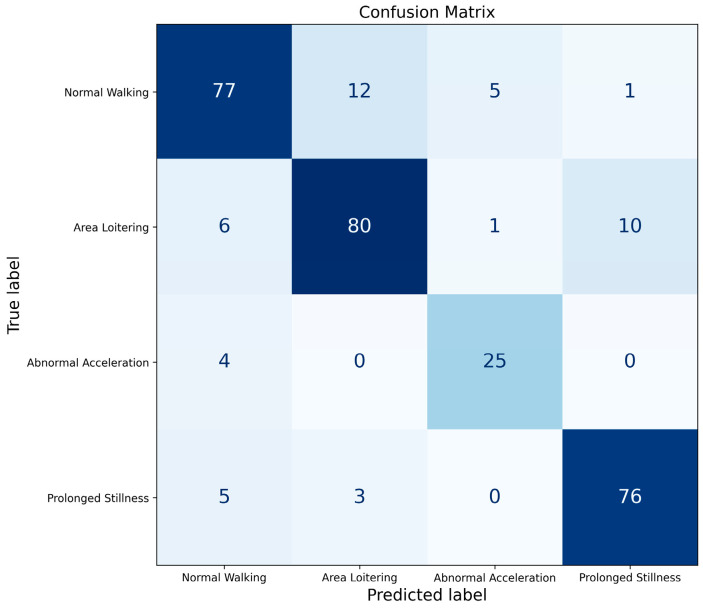
Confusion matrix, evaluating the performance of the optimized RF classifier.

**Figure 16 sensors-25-06331-f016:**
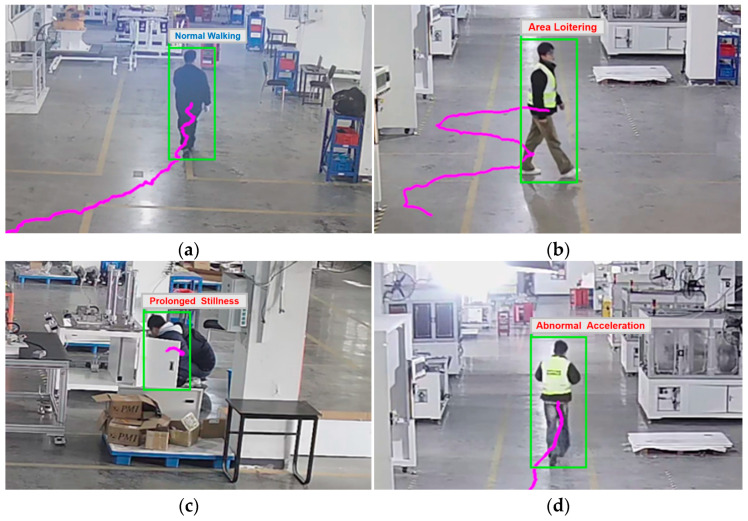
Personnel behavior recognition results: (**a**) Normal Walking; (**b**) Area Loitering; (**c**) Prolonged Stillness; (**d**) Abnormal Acceleration.

**Table 1 sensors-25-06331-t001:** Comparative experiments of different models.

Models	AP50 (%)	AP50:95 (%)	Model Size (MB)	GFLOPs	FPS
Faster-RCNN	58.3	26.7	524.5	386.2	32
YOLOv5-n	69.4	37.2	3.6	4.1	702
YOLOv5-s	72.7	38	13.7	15.8	399
YOLOv7-tiny	74.3	39.2	11.7	13	285
YOLOv8-n	71.3	40.6	5.9	8.1	433
YOLOv8-s	77.5	43.4	21.5	28.4	181
YOLOv9-t	70.9	37.8	5.8	10.7	172
YOLOv10-n	70.6	35.9	5.5	8.2	415
Ours	78.2	44.8	6	8.4	201

**Table 2 sensors-25-06331-t002:** Ablation study results.

Baseline	RFA	EMA	AP50 (%)	AP50:95 (%)	Model Size (MB)	GFLOPs	FPS
YOLOv8n			71.3	40.6	5.9	8.1	433
	√		75.8	44.2	6.0	8.3	237
		√	74.1	43.5	6.0	8.1	369
	√	√	78.2	44.8	6.0	8.4	201

**Table 3 sensors-25-06331-t003:** Environment-Specific test results.

Condition	Models	AP50(%)	AP50:95(%)
(a)	Baseline	67.7	37.4
	Ours	75.7	42.2
(b)	Baseline	60.8	29.3
	Ours	71.7	38.1
(c)	Baseline	71.3	40.6
	Ours	78.2	44.8

**Table 4 sensors-25-06331-t004:** Tracking performance of three scenes.

Model	Scene	MOTA(%)	MOTP(%)	FPS
Improved YOLOv8n + BOT-SORT	1	86.95	7.81	59.2
	2	80.10	11.84	57.3
	3	71.92	13.90	56.7

**Table 5 sensors-25-06331-t005:** Personnel behavior recognition result statistics.

	Precision (%)	Recall (%)	F1-Score (%)	Number of Test Samples
Normal Walking	83.4	81.1	82.4	95
Area Loitering	84.4	83.5	83.9	97
Abnormal Acceleration	80.6	86.2	83.3	29
Prolonged Stillness	88.4	90.5	89.4	84

## Data Availability

Data are not publicly available and can be obtained by contacting the corresponding author if necessary.

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
