# Peer review of "A Visual Trajectory-Based Method for Personnel Behavior Recognition in Industrial Scenarios"

_sensors, 2025, doi:10.3390/s25206331_

Round 1
Reviewer 1 Report
Comments and Suggestions for Authors
See attached file.

Author Response
Dear reviewer, I am extremely grateful for your valuable suggestions. I have revised the article according to your requirements. For the detailed information, please see the attachment.

Reviewer 2 Report
Comments and Suggestions for Authors
The article is interesting and analyzes personnel behavior, taking into account the interpretability of the deep learning model. However, in my opinion, it requires substantial revision before being suitable for publication, especially in the introduction and related work sections. I would suggest the following adjustments:
Abstract:
- Line 10: The gap in the literature is not very clear from the abstract. I would suggest adding a few lines to highlight this aspect.
- Line 22: I suggest enriching the abstract with details about results from a quantitative point of view and some hints about future research.
Introduction:
- Line 27: The introduction is almost entirely devoid of bibliographical references. I would suggest enriching it in this regard.
- Line 68: To better highlight the contributions of the work, the gap in the literature should be analysed and emphasised more clearly.
- Line 70: Before going into detail about the individual contributions, it would be useful to include a couple of lines summarising the contribution of the work, as it is currently somewhat fragmented.
Related Work
- Line 91: In my opinion, this section should be reworded and reorganised differently, as it currently appears to be a theoretical description of the implemented algorithms. I would suggest reducing the number of paragraphs and integrating a literature review on existing methods of personnel behavior recognition to highlight the reasons for choosing the techniques used in the work.
Methodologies
- Line 189: Before describing the methodology in detail, it would be useful to add a few lines providing a general description of the approach, to give the reader a clear overview, possibly accompanied by a simple, generic flowchart that is easier to understand before being further specified in Figure 3.
- Lines 233, 253, 261: I would suggest expanding the caption for Figures 3, 4, and 5 to make them self-explanatory.
Experiments and Analysis
- Line 570: Figure 12 is a bit confusing; I would suggest reorganising it to make it clearer for the reader.
- Line 631: To highlight the validity of the results, I would suggest that the authors integrate those relating to personnel behaviour recognition with the confusion matrix.
- Line 647: In my opinion, a more in-depth discussion of the results should be included, comparing the work with what already exists in the literature to highlight its strengths and limitations in greater detail.
Conclusions
- Line 659: I would suggest expanding and more deeply exploring the paragraph concerning future developments.
Author Response

(The authors gave the same response as above.)

Reviewer 3 Report
Comments and Suggestions for Authors# Strengths
- The research addresses a practical and critical need in industrial scenarios—personnel behavior recognition for asset protection and workplace safety—effectively filling the gap of traditional manual inspection methods (e.g., delayed response, limited coverage) in large-scale industrial environment monitoring.
- The proposed framework integrates multiple technical modules (enhanced YOLOv8, BOT-SORT, perspective transformation, and random forest) in a logical and structured manner, forming a complete workflow from personnel detection/tracking to trajectory correction and behavior recognition, which ensures the integrity and systematicity of the solution.
- Targeted improvements are made to key technical bottlenecks:
(a)- Introducing RFAConv and EMA mechanisms into YOLOv8n effectively enhances the detection accuracy of distant small targets and anti-occlusion capability, addressing the problems of missed detections and localization errors in industrial environments with complex lighting and dense equipment occlusions.
(b)- Applying perspective transformation corrects trajectory distortions caused by camera angles, which is a key innovation to improve the reliability of subsequent behavior analysis.
- The design of discriminative trajectory features (e.g., Trajectory Curvature Entropy, Primary Direction Angle) overcomes the limitation of conventional trajectory features with weak representational capability, and the validation via feature importance analysis and Pearson correlation coefficient confirms the rationality and non-redundancy of the feature set.
- The study uses a self-built industrial scene dataset (covering diverse lighting, personnel work states, and multi-person scenarios) for experiments, and adopts comprehensive evaluation metrics (AP50, AP50:95 for detection; MOTA, MOTP for tracking; Precision, Recall, F1-score for behavior recognition), ensuring the authenticity and persuasiveness of the experimental results.
# Weaknesses
- Insufficient validation of the detection model in complex industrial sub-scenarios: Although the ablation study verifies the effectiveness of RFAConv and EMA, the experiments only use a general self-built industrial dataset. There is a lack of targeted testing in extreme industrial sub-scenarios (e.g., high-temperature/high-humidity environments with severe image noise, dense overlapping of personnel and large equipment, or low-light conditions at night), making it impossible to confirm the model’s robustness in more challenging practical environments.
- Limitations in the tracking algorithm’s handling of long-term occlusion: The paper acknowledges that tracking performance degrades in Scene 3 due to extended target concealment by large obstructions, but no optimization strategies (e.g., trajectory prediction based on historical motion patterns, multi-camera fusion tracking) are proposed to mitigate this issue. This limits the applicability of the system in industrial scenes with frequent long-term occlusions (e.g., heavy machinery operation areas).
- Serious imbalance in the behavior recognition dataset and inadequate improvement measures: The Abnormal Acceleration category accounts for only 9.52% of the total trajectory samples, leading to significantly lower recognition performance (Precision=78%, Recall=72%, F1-score=75%) compared to other categories. The paper only attributes this to small sample size but does not propose solutions such as data augmentation for minority classes (e.g., synthetic trajectory generation), transfer learning, or weighted loss functions, which weakens the practicality of the system for identifying critical abnormal behaviors.
- Lack of comparative experiments with state-of-the-art (SOTA) methods:
(a)- For personnel detection, there is no comparison with other advanced small-target detection models (e.g., YOLOv9, Faster R-CNN with attention mechanisms) to demonstrate the superiority of the enhanced YOLOv8n.
(b)- For behavior recognition, there is no comparison with SOTA trajectory-based or video-based behavior recognition methods (e.g., methods using transformer architectures, 3D CNNs), making it difficult to evaluate the competitiveness of the proposed random forest-based approach.
- Inadequate analysis of computational efficiency and deployment feasibility: Although the paper mentions that the model meets lightweight deployment constraints (model size=6.0 MB, FPS>56), it does not provide specific deployment tests on edge devices commonly used in industrial scenarios (e.g., industrial control computers, edge AI boxes with limited computing power). Additionally, there is no analysis of the trade-off between model performance and computational cost (e.g., whether FPS can be maintained when the number of tracked personnel increases to 20+), which affects the assessment of the system’s practical deployment potential.
- Insufficient interpretability of the behavior recognition model: While Grad-CAM is used to visualize the detection model’s attention, there is no analysis of why the random forest classifier makes correct/incorrect judgments for specific behaviors (e.g., which features are the key factors leading to misclassifying Normal Walking as Abnormal Acceleration). This lack of interpretability is unfavorable for debugging and optimizing the model in practical industrial applications.
- Ambiguity in the perspective transformation operation: The paper states that perspective transformation relies on manual annotation of four vertices of the rectification area (e.g., rectangular ground markers), but it does not explain how to handle industrial scenes without obvious markers or how to achieve automatic calibration of feature points. Manual annotation increases the system’s deployment cost and reduces its scalability.
Author Response

(The authors gave the same response as above.)

Round 2
Reviewer 1 Report
Comments and Suggestions for Authors
The paper notably improved after its revision, and I have no further comments.
Reviewer 2 Report
Comments and Suggestions for Authors
The authors have addressed all the concerns I raised, improving the quality of their work. Therefore, I recommend accepting the manuscript in the current version.
Reviewer 3 Report
Comments and Suggestions for Authors
The revised paper can be accepted